# Sub-100-fs energy transfer in coenzyme NADH is a coherent process assisted by a charge-transfer state

Vishal Kumar Jaiswal [1], Daniel Aranda Ruiz [2,6], Vasilis Petropoulos [3,6], Piotr Kabaciński [3,6], Francesco Montorsi[1], Lorenzo Uboldi [3], Simone Ugolini [1], Shaul Mukamel [4], Giulio Cerullo [3] ✉, Marco Garavelli [1] ✉, Fabrizio Santoro [5] & Artur Nenov [1] ✉

Excitation energy transfer (EET) is a key photoinduced process in biological chromophoric assemblies. Here we investigate the factors which can drive EET into efficient ultrafast sub-ps regimes. We demonstrate how a coherent transport of electronic population could facilitate this in water solvated NADH coenzyme and uncover the role of an intermediate dark charge-transfer state. High temporal resolution ultrafast optical spectroscopy gives a 54±11 fs time constant for the EET process. Nonadiabatic quantum dynamical simulations computed through the time-evolution of multidimensional wavepackets suggest that the population transfer is mediated by photoexcited molecular vibrations due to strong coupling between the electronic states. The polar aqueous solvent environment leads to the active participation of a dark charge transfer state, accelerating the vibronically coherent EET process in favorably stacked conformers and solvent cavities. Our work demonstrates how the interplay of structural and environmental factors leads to diverse pathways for the EET process in flexible heterodimers and provides general insights relevant for coherent EET processes in stacked multichromophoric aggregates like DNA strands.

Following the excitation energy transfer (EET) between chromophoric units in multimeric aggregates is of key importance for understanding biological processes like photosynthesis and DNA photodamage, as well as for the design of efficient light-harvesting molecular assemblies.[1–3] EET in these systems falls between the regimes of coherent and incoherent energy transport.[4] Where a process is placed in this sliding scale is dictated by: a) the energetic separation of relevant electronic states versus the coupling strength to each other and

with the environment[5–9]; b) the spatial separation of the monomers in the macro-structure.

In homoaggregates, coherent population oscillations of quantum origin between the individual sites coupled to each other with a fixed phase relation in so-called excitonic excited states (ES) arise due to purely electronic coupling between the monomers. Coupling the electronic dynamics to a bath of nuclear degrees of freedom facilitates an efficient and ultrafast unidirectional population transfer to the

[1]Dipartimento di Chimica industriale "Toso Montanari", Università di Bologna, Viale del Risorgimento 4, 40136 Bologna, Italy. [2]ICMol, Universidad de Valencia, Catedrático José Beltrán Martínez, 2, 46980 Paterna, Spain. [3]Dipartimento di Fisica, Politecnico di Milano, Piazza Leonardo da Vinci 32, 20133 Milano, Italy. [4]Department of Chemistry and Department of Physics and Astronomy, University of California, Irvine, CA 92697, USA. [5]Istituto di Chimica dei Composti Organometallici (ICCOM-CNR), Area della Ricerca del CNR, Via Moruzzi 1, I-56124 Pisa, Italy. [6]These authors contributed equally: Daniel Aranda Ruiz, Vasilis Petropoulos, Piotr Kabaciński. ✉e-mail: giulio.cerullo@polimi.it; marco.garavelli@unibo.it; artur.nenov@unibo.it

lowest excitonic state. The role of such electronic coherences in disordered biological systems, such as the reaction centers of light-harvesting complexes, is being intensely debated[10–13].

In heterodimers with energetically well-separated electronic states, the motion of the nuclear wavepacket can strongly tune the energy gaps promoting crossings where electronic and vibronic couplings, i.e. wave function mixing along nuclear degrees of freedom, facilitate the population transfer[14–21]. In the limit of large donor-acceptor separation (i.e. >10 Å) inter-chromophore orbital overlaps are negligible, and couplings are dictated by the long-range electrostatic interactions. As the resulting electronic couplings are weaker than the coupling to the environment, the coherently prepared excited state first thermalizes through vibrational cooling on the picosecond time scale[22,23]. The EET then occurs incoherently on tens of ps-to-ns time scale, with rates that can be estimated by utilizing Förster resonance energy transfer theory. In the case of closely packed hetero-aggregates, on the other hand, variations of the inter-chromophore orbital overlaps and mixing with charge-transfer (CT) configurations[24,25] occurring during the photoinduced vibrational dynamics give rise to electronic and vibronic couplings which can be as pronounced as their intra-molecular counterparts, accelerating the EET down to the sub-ps regime. In this limit, the coherent motion of the nuclear wavepacket before thermalization (referred to as classical coherence) induces an inherently coherent EET competitive even with possible sub-ps decay channels such as intramolecular internal conversion (IC) to the ground state (GS) mediated by conical intersections (CIs).

NADH, the reduced form of nicotinamide adenine dinucleotide (NAD), is an important coenzyme found in the mitochondria that is part of the electron transport chain leading to the generation of adenosine triphosphate[26]. It is a dimer consisting of the two chromophores adenine (Ade) and nicotinamide (Nic), which absorb light in the ultraviolet (UV) region, connected through a phosphate bridge. This bichromophoric structure provides a playground rich with photoinduced processes, which are used as spectroscopic markers for cellular metabolism, for example, in label-free multimodal microscopy[27,28].

The high conformational flexibility afforded by the phosphate bridge linking the two monomers leads to DNA-like stacked (folded) or unstacked (unfolded) conformers whose ratio depends on solvent polarity, pH, and temperature[29–33]. While in alcoholic environments, like propylene glycol or methanol, there is a high propensity of unstacked conformations, a considerable percentage of stacked conformers is observed in water[34–37]. In its folded state, NADH is characterized by a photoinduced ultrafast EET process from Ade (donor) to Nic (acceptor). Since its first observation by Weber in 1957[34], the EET process, which involves the frontier π-orbitals of the locally excited (LE) states of the two monomers[38,39], has been extensively studied. Upon impulsive resonant excitation of the $L_a$-state of Ade, immediate (within the ≈100-fs temporal resolution of the ultrafast experiments) emission was observed from the lowest ππ* state of Nic (referred henceforth as Nic*). Previous time-resolved fluorescence and transient absorption (TA) experiments revealed the ultrafast nature of the process, placing the EET timescale in the sub-100-fs range[36,37], but with significant uncertainty due to their limited temporal resolution.

In this letter, using the example of NADH, we illuminate the structural, energetic and environmental conditions which favor a sub-100-fs coherent EET mechanism against IC in a closely stacked molecular heteroaggregate. Using UV TA spectroscopy with sub-30-fs temporal resolution, we resolve the EET from Ade to Nic* in water and find that, with a time constant $\tau_{EET} = 54 \pm 11$ fs, it is much faster than intra-Ade IC to the GS, which occurs with a time constant $\tau_{IC} \sim 160$ fs[40–46]. This observation places the mechanism through which the EET occurs completely outside the applicability of Förster theory still employed in the analysis of NADH photophysics[36,37]. Through multidimensional nonadiabatic quantum dynamics, we anticipate that

this ultrafast EET in folded conformers is channeled through coherent vibrational motion along the same planar modes that facilitate intra-Ade IC. These coherent molecular vibrations contribute to the EET being faster than IC to the GS as, for the latter process, additional out-of-plane structural deformations are needed to bring the $L_a$ and GS electronic potential energy surfaces to degeneracy. This is a general system-independent observation which implies that EET is a competitive deactivation process in closely stacked aggregates. Furthermore, our simulations reveal a dark Nic→Ade CT state, highly susceptible to structural and environmental disorder. In favorably stacked dimers and in cooperation with the fluctuations of the polar environment, the CT state is stabilized in the vicinity of the $L_a$ state and acts as an intermediary that further boosts the EET yield.

## Results

### Transient absorption spectra of adenosine and NADH

Figure 1a, b depict the TA maps of Adenosine (a) and NADH (b) in water upon resonant excitation of the Ade moiety at 4.7 eV with sub-20-fs pulses and broadband (1.9-3.2 eV) probing. The excited state of the Ade monomer exhibits an intense photo-induced absorption (PA) band that spans our entire probe photon energy range, characterized by a sharp peak at >3 eV (Fig. 1a). This PA band decays on the <300-fs timescale as a result of Ade's $L_a$-state IC pathway back to the ground state. This is consistent with the vibronically driven 100–300 fs IC, reported for Ade and its derivatives[40–47].

The 4.7 eV UV pulse predominantly photoexcites the Ade moiety in NADH, resulting in an initial PA band similar to the one reported for the monomeric Ade (Fig. 2b). As mentioned earlier, the equilibrium between folded/unfolded conformers in NADH heavily depends on the solvent. In methanol, NADH exists almost entirely in its open form and thus the EET pathway is negligible, resulting in excited state dynamics similar to isolated Ade in water (see Suppl. Fig. 1 and Suppl. Fig. 2 in the Supplementary Information (SI)). However, in water, NADH has an approximate 30/70 population ratio between folded and unfolded configurations[37]. The significant population of folded structures activates the EET from Ade to Nic, resulting in an additional ultrafast deactivation channel of Ade's excited state (Fig. 1b). At 3.1 eV probe photon energy, the EET process results in a long-lived signal, due to a PA band from the Nic* state (Fig. 1c). In contrast, for a 2.5 eV probe, only Ade displays a significant PA. Comparison of the dynamics of Adenosine and NADH in water at this probe photon energy (Fig. 1d) reveals a faster decay for NADH, due to the additional deactivation channel for the $L_a$ excited state of Ade through EET to Nic*.

### Estimation of experimental EET timescale

Global fitting of the TA data, as shown in Table 1, gives time constants of $\tau_{1Ade} = 157 \pm 4$ fs and $\tau_{1NADH, water} = 122 \pm 4$ fs for Ade and NADH in water, respectively. While in isolated Ade and NADH solvated in methanol, the corresponding time constants of 157 fs and 167 fs correspond to the IC back to the GS, in NADH solvated in water the EET mechanism opens up an additional channel of excited state deactivation, increasing its rate. Taking into account the 30/70 equilibrium between folded and unfolded forms, the faster lifetime $\tau_{1NADH, water} = 122$ fs extracted by global analysis can be considered as a weighted average of the folded and unfolded ultrafast lifetime components:

$$\tau_{1NADH, water} = 0.7\,\tau_{1NADH, unfolded} + 0.3\,\tau_{1NADH, folded}$$

Assuming similar excited-state deactivations in unfolded water-solvated NADH and isolated Ade, one can take $\tau_{NADH, unfolded} = \tau_{1Ade, water} = 157$ fs, which results in a time constant $\tau_{NADH, folded} = 40 \pm 6$ fs for the folded NADH conformers. This total decay rate obtained for the folded NADH components ($k_{total} = (40\ fs)^{-1}$) reflects

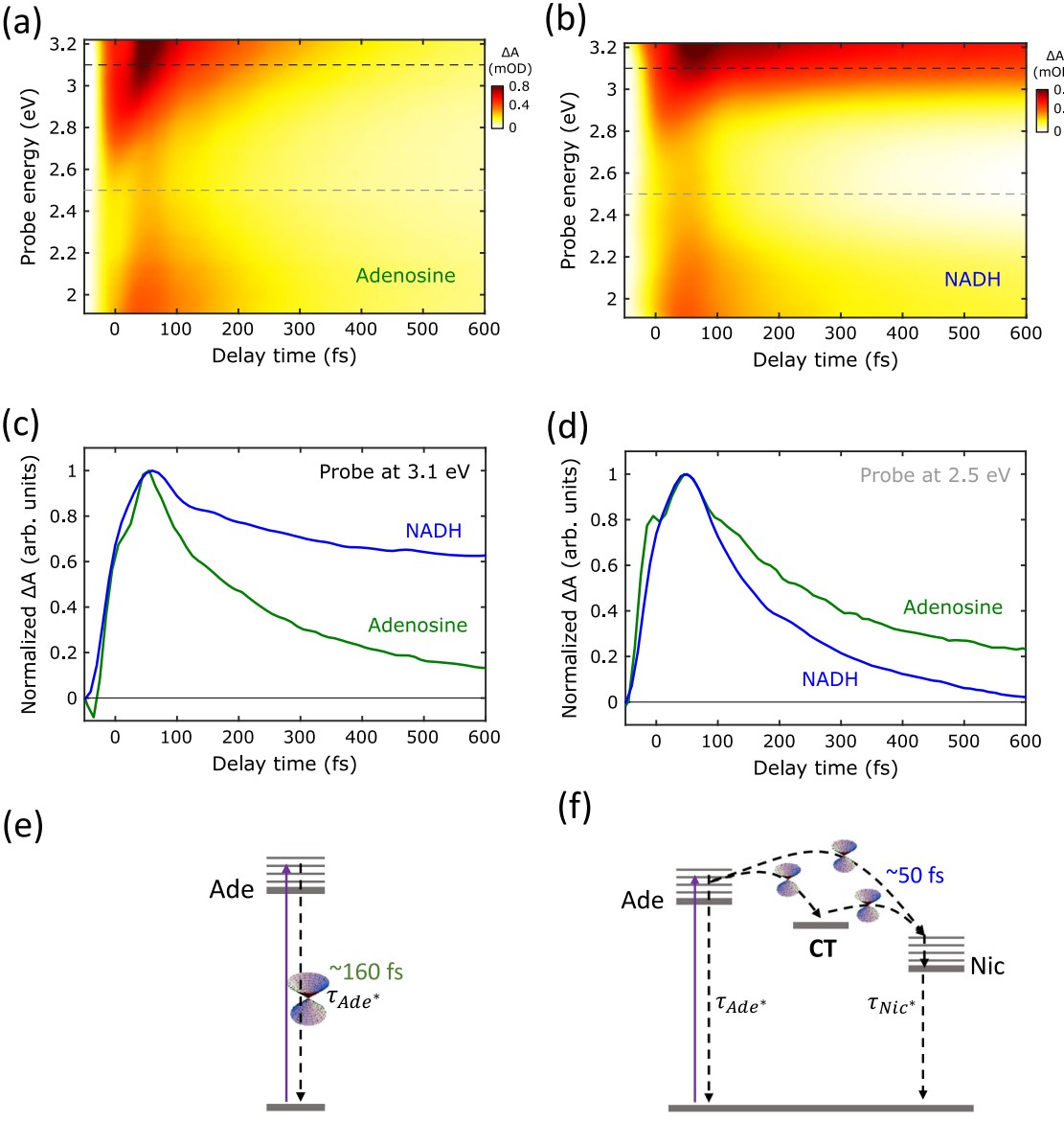

**Fig. 1 | Ultrafast energy transfer in NADH.** Transient absorption (TA) maps of **a** Adenosine (Ade) and **b** NADH in piperazine-N,N'-bis(2-ethanesulfonic acid) (PIPES) aqueous buffer upon photoexcitation with sub-20-fs pulses at 4.7 eV. The dashed lines indicate the selected time-traces shown in Fig. 1c, d for detection at 3.1 eV (black dashed line) and 2.5 eV (gray dashed line). **c, d** Time-traces of Ade (green) and NADH (blue) for probing at **c** 3.1 eV and **d** 2.5 eV respectively. The faint gray line in Fig. 1c, d defines the zero line (ΔA = 0). Schematics depicting the ultrafast vibronic processes after Ade resonant excitation which lead to **e** internal conversion (IC) in Ade and **f** direct/through-CT coherent EET in NADH. Source data are provided as a Source Data file.

the sum of the rates for Ade IC ($k_{IC} = (157 \, fs)^{-1}$) and ultrafast EET from the Ade $L_a$-state to Nic* ($k_{ET}$), resulting in $k_{total} = k_{IC} + k_{ET}$, thus giving a time constant of $\tau_{ET, \, folded} = 54 \pm 11 \, fs$ for the EET process (see a detailed discussion in Suppl. Note 1.2 and 1.3).

**Quantum dynamics simulations of the EET in NADH**

To gain insight into the ~50-fs Ade →Nic EET mechanism in the folded NADH forms, we have modeled the photoinduced time-evolution of the system by nonadiabatic quantum dynamics of multidimensional wavepackets using the MCTDH method[48–52]. Due to the large number of normal modes in the bichromophoric molecule, the multilayer formulation of the MCTDH method (ML-MCTDH) was employed[53–56]. The coupled dynamics of the electronic states of the system through the vibrational degrees of freedom are described by a Linear Vibronic Coupling (LVC) Hamiltonian[57,58], parametrized with the energies and gradients computed at the multiconfigurational wavefunction-based XMS-CASPT2 level of theory[59]. Calculations were performed on

different water solvated stacked NADH aggregates, obtained from a 20 ns of Replica Exchange Molecular Dynamics (REMD)[60], in a quantum mechanics/molecular mechanics (QM/MM)[61] setup allowing to account for the conformational diversity and solvent effects with quantitative accuracy. Details are provided in the Suppl. Note 2 to 7.

The structural flexibility of NADH leads to diverse relative orientations of the two bases in the stacked conformers, whose relative populations in REMD dynamics are detailed in Suppl. Table 2. Figure 2d–f shows the time-evolution of electronic states population in representative structures of the three most populated stacked conformers, capturing 65% of stacked population. The dynamics reveal that EET is an ultrafast process mediated by coherent vibrational motions. The coupling between the electronic states induced by nuclear motions along these vibrations facilitates a near complete Ade →Nic population transfer on a sub-100-fs time scale. As shown in Fig. 2 and additional dynamics reported in Suppl. Note 8 of the SI, the EET mechanism can be either direct (Fig. 2f) or proceed involving a

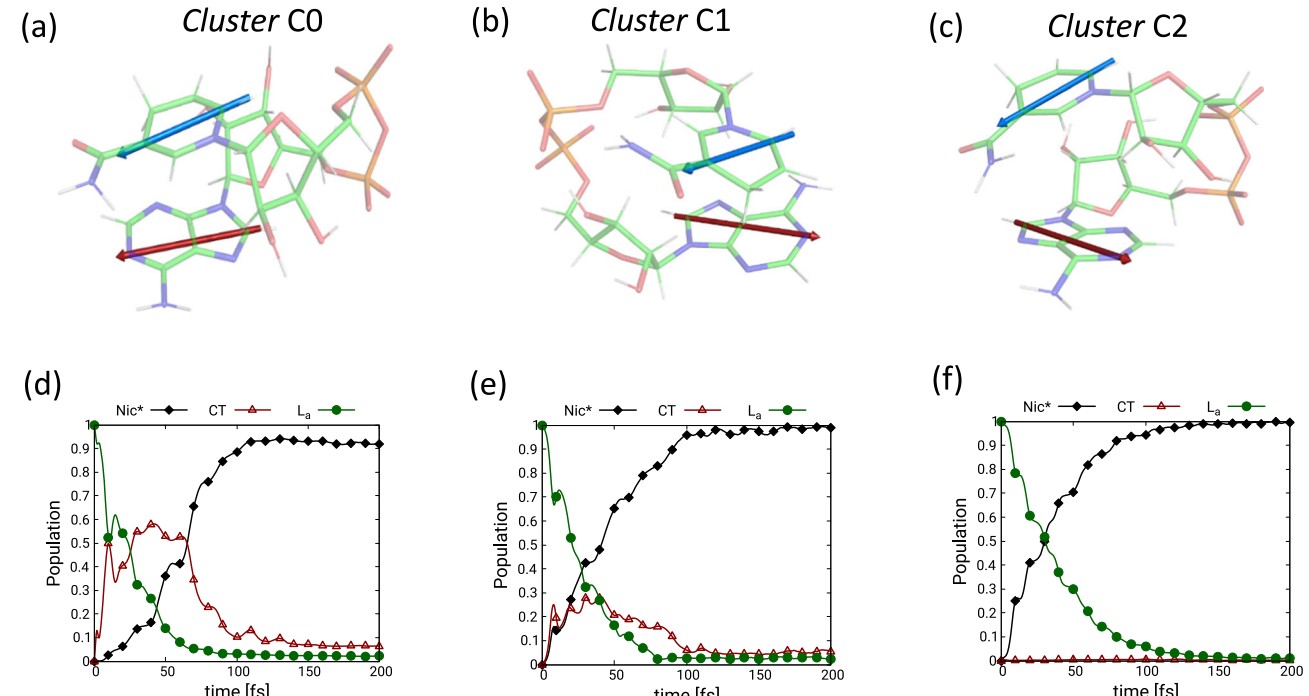

**Fig. 2 | Ultrafast energy transfer dynamics in highest populated clusters.**
**a**–**c** Conformational representatives of the three largest clusters of Replica-Exchange Molecular Dynamics. The relative orientation of two chromophores in different clusters is shown through arrows colored blue (on Nic) and red (on Ade). Carbon atoms are colored in green, nitrogens in blue, hydrogens in white, oxygen in red and phosphorus in yellow. Details about clustering and the population of clusters are given in Suppl. Note. 2. **d**–**f** Quantum Dynamics (using ML-MCTDH) on a representative structure of the three largest clusters. Additional dynamics on other representative structures are shown in Suppl. Fig. 10. Source data are provided as a Source Data file.

**Table 1 | Time constants, accompanied by their experimental uncertainty, as obtained from the global fitting of the TA data in the case of Adenosine in water and NADH in water**

|  | $\tau_1$ (fs) | $\tau_2$ (fs) | $\tau_3$ (fs) |
|---|---|---|---|
| **Adenosine in water** | 157 ± 4 | 894 ± 50 | - |
| **NADH in water** | 122 ± 4 | 878 ± 40 | long |

Nic→Ade CT state as an intermediary (Fig. 2d,e). The propensity for a direct or CT-mediated EET mechanism is dictated by the energetic position of the CT state versus the $L_a$. We shall show later how this is affected by the structural and solvent heterogeneity present in the system.

Vibronically coherent mechanisms are ubiquitous in ultrafast photophysical processes involving population transfer between electronic states[62,63]. Impulsively excited molecular vibrations, called tuning modes, modulate the energy gaps between electronic states, whereas vibrations known as coupling modes, together with purely electronic coupling, induce wave function mixing which dictates the rate of the non-adiabatic population transfer between electronic states. The direct transfer in NADH is mediated by high-frequency stretching modes of the initially excited Ade moiety. In particular, the C-C and C-N stretchings, with frequencies of ~1400 cm$^{-1}$ and ~1600 cm$^{-1}$, respectively, show the highest tuning and, simultaneously, significant coupling between $L_a$ and Nic*, irrespective of stacking conformation (Suppl. Fig. 15 in the SI). The oscillatory pattern in the first 100 fs of the population dynamics (Fig. 2f) with a period of ≈10 fs is the result of the tuning modes leading to energy degeneracy (real $L_a$/Nic* CI) between $L_a$ and Nic* every half a period (Suppl. Fig. 12 and Suppl. Fig. 16 in SI).

To gain further insight, we computed the quantum electronic coherences between $L_a$ and Nic* along the dynamics (Suppl. Note. 15)

and observed that they last for at least 70 fs while exhibiting recurrences with a period comparable to the frequencies of the stretching modes (Suppl. Fig. 18). Their eventual decay is caused by the concurrent effects of: a) progressive population depletion in the $L_a$ state accompanied by the accumulation in the Nic* state; b) progressive decrease of the overlap of the wavepackets evolving in the $L_a$ and Nic* states. The direct EET is facilitated to equal amounts by electronic ($E_{ij}^0$) and vibronic ($\lambda_{ij}$) couplings, which can be verified by switching off their respective contribution (Suppl. Fig. 18). The population transfer becomes slower in both cases, yet, notably, population inversion still occurs within 100 fs. Interestingly, the observation of ultrafast EET in the absence of electronic couplings makes apparent that in closely stacked aggregates even small localized deformations such as stretchings induce vibronic couplings large enough to facilitate a sub-100-fs EET.

We note that intra-monomer IC in Ade is initially driven by the same impulsively excited high-frequency modes. However, as the energy gap (with the GS) to overcome is significantly bigger (ca. 5 eV with respect to ca. 1.5 eV), additional deformations, such as ring puckering, are necessary to facilitate IC to the GS[42,46,64,65]. Since these deformations need activation and are associated with more pronounced displacements, IC is naturally slower, taking a few hundreds of fs. Source data are provided as a Source Data file.

### Factors affecting the participation of CT state in EET process
The extent of participation of the dark CT state in the EET process is determined by its energetic position and coupling to the two monomer LE states $L_a$ and Nic*. The structural and solvent dynamics impart a large amount of heterogeneity to the system, leading to various types of stacked conformers with different relative orientations of the two bases (Fig. 2a–c). While the energetic positions of the $L_a$ and Nic* are relatively unaffected by the different arrangements of the two chromophores, the CT state is markedly more sensitive. Figure 3 shows how

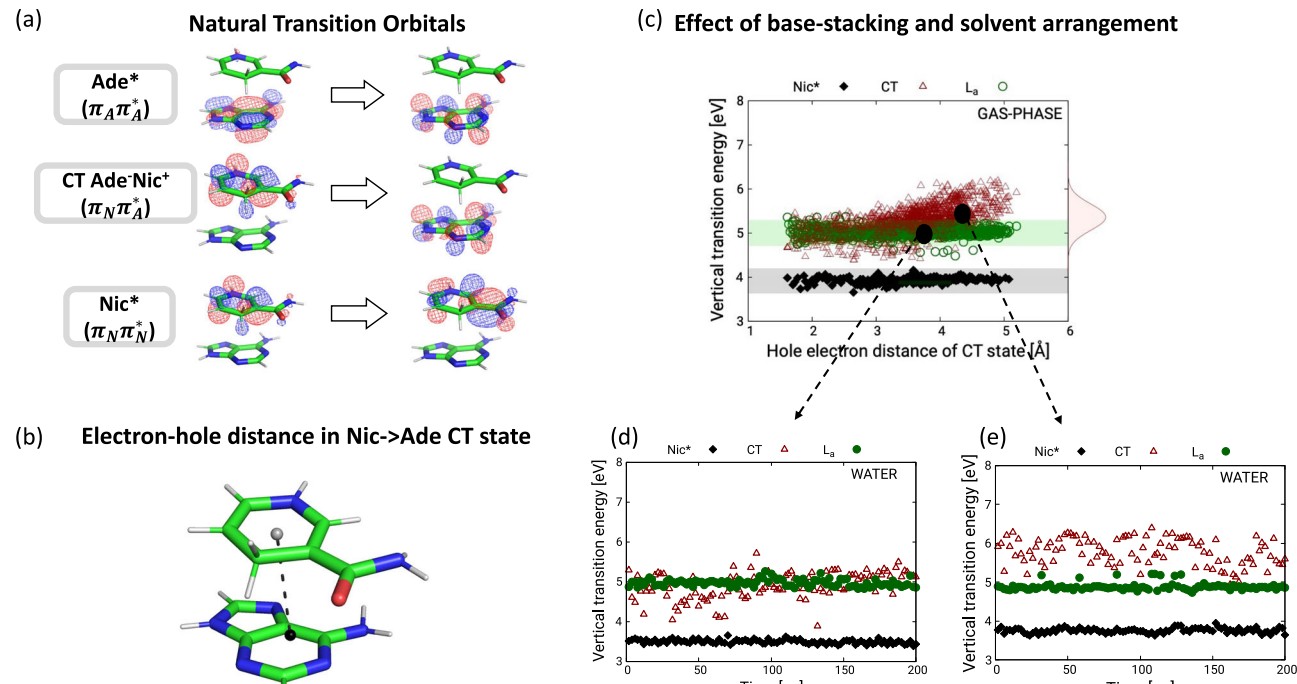

**Fig. 3 | Stacking and solvent effects on energies of charge-transfer (CT) and locally excited (LE) states. a** The natural transition orbitals of the two LE and lowest CT state involved in the EET process. **b** The electron-hole distance[77,78] in the Nic→Ade CT state in a representative stacked conformer depicted with a dashed line. The two dots specify the position of the center of charges of hole and electron created in the CT state. **c** Stacking Effect: Vertical transition energies of the CT state vs the electron-hole distance of the CT state in gas-phase. The black and green bands indicate the energies of Nic* and $L_a$ state which show minimal fluctuation compared to CT state. The Gaussian profile shows the spread of the values of the CT state with FWHM of ca. 1 eV. Two structures with markedly different CT energies are highlighted with black dots for studying solvent effects. **d, e** Solvent Effect: The effect of the solvent fluctuations on the energies of the diabatic states for the two structures highlighted in (**c**), which represent respectively a case in which the CT is near degenerate with the $L_a$-state of Ade (**d**) and a case in which the CT state is significantly higher in energy (**e**). The two profiles were obtained by running an equilibrated solvent MD around the two frozen NADH solute geometry and computing vertical transition energies at XMS-CASPT2 level. Details in Suppl. Note. 14.

both the stacking and solvent configuration individually affect the energetic position of the CT state. The transfer of an electron from Nic to Ade in the CT state creates an electron-hole dipole in the system (Fig. 3b). The relative orientation of the two bases modulates the distance between the centers of (positive and negative) charge of this electron-hole pair in stacked conformations from 3 to 5 Å. In return, this causes a variation of the CT state energy by ~1 eV in vacuo (Fig. 3c) at the time of photoexcitation. Introducing the solvent, the interaction of the CT dipole with the electric field of water additionally modulates the fluctuation of CT state energy by ~1 eV, as seen from computations in solvent ensemble around two geometries selected from the manifold of stacked conformations (Fig. 3d, e).

The combined effects of stacking and solvent orientation dictate whether the CT state can actively participate in ultrafast EET process. If a certain stacking places the CT state more than 0.5 eV above the $L_a$ state in gas-phase, its energy fluctuations due to the interaction with the solvent cannot render the CT accessible from the $L_a$ state after photoexcitation (Fig. 3e). In such conformations, populating predominantly cluster C2 – with a median interbase distance of ~4.6 Å – the EET occurs via a direct transfer. Instead, for stacking where the CT state is energetically closer to the $L_a$ (i.e. within less than 0.5 eV) in gas-phase, the solvent fluctuations can induce stabilization of the CT band around and even below the $L_a$ state (Fig. 3d), eventually enabling its active participation in the EET dynamics. Such is the case for the most populated stacked cluster C0 – with a median interbase distance of ~4.12 Å. The average ratio between La/CT electronic coupling and energy gap $|V/\Delta| = 0.268$ calculated for the conformations in this cluster indicates a strong wave function mixing and, consequently, admixing of CT character in the bright adiabatic state in the Franck-Condon (FC) region.

Figure 2d shows a representative dynamics for a snapshot of cluster C0 in which the CT state is 0.5 eV below the $L_a$ in the FC region. A virtually immediate increase of the population of the CT state to 50% on a 10 fs timescale is observed, which is accompanied by the instantaneous formation of a quantum coherence which lasts for about 70 fs (column LVC in Suppl. Fig. 19). Tuning along the high-frequency C-C and C-N stretching modes quickly overcomes the La/CT energy gap, so that electronic and vibronic couplings – exhibiting larger magnitudes compared to their counterparts in the direct EET due to the one-electron nature of the process – efficiently promote La→CT population transfer. Notably, if tuning modes are switched off, thus forcing purely electronic dynamics between stationary wavepackets, no population transfer takes place despite the formation of coherences of sizeable magnitude (column "no $\lambda_{ii}$ All" in Suppl. Fig. 19). This makes evident that tuning modes are essential to circumvent the La/CT energy gap for the electronic and vibronic couplings to have effect. Furthermore, we selectively switched off either coupling to investigate its impact on the Nic→Ade electron transfer event (columns "no $E^0_{ij}$ All" and "no $\lambda_{ij}$ All" in Suppl. Fig. 19). Purely electronic couplings are found to promote La/CT population inversion on a 10-fs timescale. Electronic coherence formation is vibrationally assisted as can be noted by the coherence recurrences with times compatible with the periods of the high-frequency tuning modes, and by the fact that the coherence magnitude moderately increases when switching off electronic couplings.

The population of the CT state is only transient and is transferred to Nic* creating, at the same time, an electronic coherence between CT and Nic* that lasts till 150 fs. Contrary to the high frequency modes driving the direct EET, we find that multiple conformation-dependent low frequency modes delocalized over both moieties facilitate this second step of the CT-mediated EET. These low-frequency modes

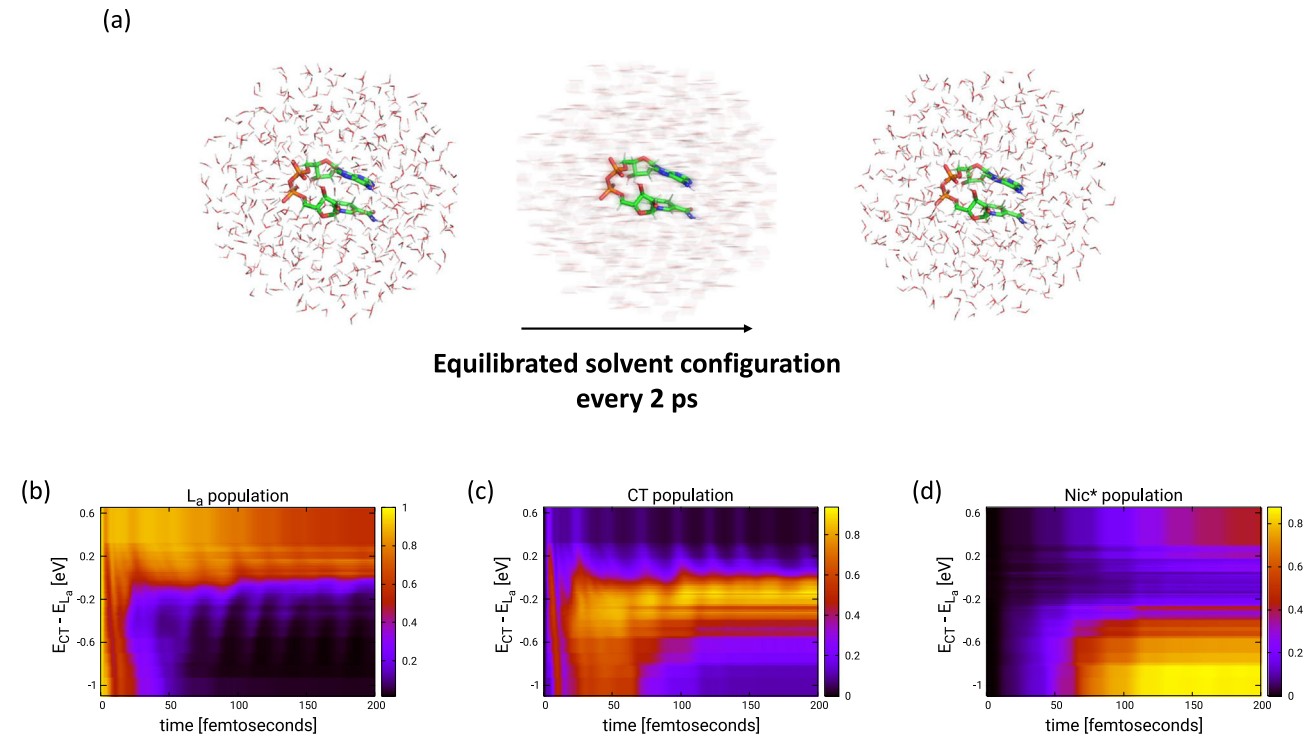

**Fig. 4 | Wavepacket dynamics on most populated conformer in 100 equili-brated solvent ensembles. a** An ensemble of equilibrated solvent configurations is created around a representative structure of most populated cluster of the REMD. The solute NADH is kept frozen during the solvent equilibrium dynamics. **b−d** Populations dynamics in the three diabatic electronic states - $L_a$ (left), CT (middle), Nic* (right) – during the first 200 fs after photoexcitation of the $L_a$ state as a function of the relative energy difference between the CT and $L_a$ states (i.e. $E_{CT}$-$E_{L_a}$). The plots are generated from individual ML-MCTDH dynamics performed on 100 snapshots from the solvent equilibrium dynamics as indicated in (**a**), each one with characteristic solvent-modulated electronic energies, electronic and vibronic couplings. Details on the simulation can be found in Suppl. Note. 14. Source data are provided as a Source Data file.

show the largest vibronic coupling between the CT and Nic* state amongst the normal modes of the system. When the vibronic couplings along these modes are switched off (see Suppl. Fig. 14 in SI for a representative case), the transfer to Nic* from the CT state is significantly slowed down and the population remains trapped in the CT state.

To explore the role of solvent heterogeneity in the ultrafast EET process, we carried out independent wavepacket dynamics on a representative structure from the most populated conformer embedded in 100 different solvent configurations obtained through equilibrated molecular dynamics at room temperature (Fig. 4a). Thereby, for each solvent arrangement the LVC Hamiltonian was reparametrized to take into account the solvent effect on the energetics and electronic coupling (details in Suppl. Note. 14.1). Figure 4b shows a false color plot of the state-specific population evolution upon instantaneous $L_a$ photoexcitation along the 100 wavepacket dynamics ordered according to the energy gap between CT and $L_a$ in the FC point. An ultrafast population transfer to Nic* is observed when the CT state is energetically below the $L_a$ state at the FC geometry. The CT state acts as a doorway populated within 10-20 fs after excitation of the $L_a$ state and depopulated within 60 fs by transferring the population to the Nic* state. The acceleration provided by the CT state becomes apparent when comparing against dynamics performed in the same solvent ensemble after switching off LE-CT couplings (Suppl. Fig. 17 in SI, also see Suppl. Figs. 10/S11) where the EET process gets substantially slowed down. A similar retardation is observed for those solvent/stacking configurations where the CT state is substantially above the $L_a$ (Fig. 4b) and, therefore, cannot act as a doorway in the EET mechanism. Thus, the acceleration provided by the CT state can play a paramount role in making the EET mechanism the dominant relaxation path immediately following photoexcitation.

Besides the ultrafast population transfer to Nic*, we observe population trapping in the CT state in our dynamics reported in Fig. 4 when $L_a$ and CT are energetically close in the FC region. In this case, after an ultrafast population of the CT state, a subsequent transfer to Nic* state is substantially slowed down as the CT and Nic* surfaces do not cross in the course of the vibrational dynamics in the CT state. However, it should be noted that our model lacks the dynamic relaxation of the electronic states induced by the temporal response of the polar solvent. Indeed, a charge-separated state can access lower energies upon relaxation of the molecules of the surrounding polar solvent which can occur on a timescale as fast as sub-50-fs in water[66–68]. Thus, we expect that the CT state stabilization induced by modeling the response of the polar solvent will further put these dynamics into the realm of ultrafast population transfer to Nic* state.

## Discussion

In this work, we unambiguously demonstrate, by means of TA spectroscopy with sub-30-fs temporal resolution, that the ultrafast Ade to Nic EET in water-solvated NADH occurs on a 50-fs timescale. By means of multimode wavepacket quantum dynamics, we reveal that this is an electronically and vibrationally coherent process. Coherent molecular vibrations – high-frequency stretchings – activated upon optical excitation of Ade (the donor) tune the energy gap where electronic and vibronic couplings directly channel the electronic population to Nic (the acceptor). Depending on the spatial separation and relative orientation of the two monomers (which affects the magnitude of the couplings) the direct pathway exhibits timescales ranging from few tens of fs to several ps. Sub-100 fs direct EET pathways are only possible for very close (< 5 Å) stacking, since this activates inter-chromophore vibronic couplings of the same order of magnitude as intra-chromophore ones.

The EET process can be further accelerated via the mediation of a low-lying CT state coupled to the locally excited states of the two monomers. The relative orientation of the two monomers in closely stacked conformations, along with the solvent thermal fluctuations, affect the energetics of CT state and determine the propensity of the CT-mediated EET pathway. When the CT state is near-degenerate or below the $L_a$ state of Ade, the first step of the CT-mediated EET, i.e. the Ade→Nic hole transfer, occurs quasi instantaneously. Completion of the EET process is observed within 100 fs, through the subsequent Ade→Nic electron transfer, which is dependent on the relaxation timescale of the solvent.

Our findings are of general validity for describing the dependence of energy transfer and charge separation processes on the coupling between electronic and nuclear degrees of freedom, conformational heterogeneity and solvent fluctuations in closely packed hetero-aggregates. The EET depends on the interplay of many factors: nature of the tuning and coupling modes and respectively the amplitude of their motion or the strength of coupling they carry; conformational freedom and spatial distance between chromophores in the aggregate; solvent polarity and thermalizaiton timescale. We identify the parameter window which facilitates ultrafast EET. At an inter-chromophore distance of <5 Å: a) electronic and vibronic couplings are equally relevant and cooperate in promoting efficient EET; b) even localized vibrational modes are capable of generating sufficiently strong orbital mixing and, thus, strong vibronic couplings; c) the short distance between the centers of negative and positive charge of the two chromophores stabilizes the CT state opening an additional EET pathway, favored in polar solvents. Most importantly, despite the conformational heterogeneity of such macro-structures, the EET is an intrinsically coherent process, i.e. governed by coherently oscillating wavepackets on the potential energy surfaces of the involved electronic states which tune the energy gaps and drive the EET unidirectionally from the donor to the acceptor. Such coherent EET cannot be described by Förster theory, which is suited to describe incoherent EET mechanisms.

Classical coherences manifest themselves through quantum beating in the TA spectra[69–71], yet, the high frequencies of the dominant modes in NADH prevent them from being registered with current state-of-the-art experimental setups. This makes the theoretical analysis much more relevant. We hope that our findings will be a stimulus to increase the temporal resolution of UV TA spectroscopy and to apply transient spectroscopies which interrogate other spectral ranges in which the CT state may show unambiguous fingerprints. In particular, we propose photoelectron spectroscopy as a more sensitive way to single out CT states, as ionization from the intermediately created negatively charged Ade would give rise to characteristic signatures at lower ionization energies compared to the neutral form. Eventually, this study discloses a learning path on how to underdstand, and thereafter tune, the control knobs that are crucial for the design of systems with tailored EET efficiency.

## Methods

### Experimental setup

Ultrafast TA experiments[72] were performed using a Ti:sapphire laser generating 100 fs pulses at 800 nm wavelength and 1 kHz repetition rate. Deep UV pump pulses tunable in the 4.2-4.8 eV range were generated as the second harmonic of a visible non-collinear optical parametric amplifier and compressed to sub-20-fs duration with a prism pair. Probe pulses covering 1.9-3.2 eV were obtained through white-light continuum generation by focusing a fraction of the fundamental beam in a calcium fluoride plate. Pump and probe polarizations were set at the magic angle (54.7°).

### Sample preparation

β-NADH, in the form of reduced disodium salt hydrate, and adenosine were purchased from Sigma-Aldrich and used as received, dissolved in either methanol or 0.1 M piperazine-N,N′-bis(2-ethanesulfonic acid) (PIPES) aqueous buffer at pH 7.0. The samples were flown in a 150-μm-thick laminar liquid jet configuration, resulting in the absorbance of 2 OD at 4.7 eV pump photon energy. The used pump fluence was below 300 μJ/cm² to minimize the coherent artifact and solvated electron signals.

### Estimation of experimental EET timescale

The timescale of the EET process was estimated by employing a sequential kinetic model, taking into account the diverse relaxation pathways undertaken in stacked and unstacked conformers. The timescale of the EET pathway was estimated from the obtained lifetimes of the various evolution-associated spectra (EAS) obtained through global analysis of the TA map. The full details are reported in Suppl. Note. 1.3.

### Computational methods

The conformational heterogeneity of solvated NADH was sampled through replica-exchange molecular dynamics performed through AMBER molecular dynamics package as detailed in Suppl. Note. 2. The major conformers were identified through a cluster analysis using of internal distance matrix of the solute atoms. The population fraction and inter-base distances of major clusters are reported in Suppl Table 2. Hybrid QM/MM calculations were carried out with the COBRAMM[73,74] program, interfacing Gaussian16[75] and openMOLCAS[76] QM codes with the AMBER molecular dynamics package. The QM and MM partitioning involves three layers: High, Medium, and Low. The two chromophore bases in NADH solute are treated at QM level (High level), while the rest of the backbone along with the solvent is treated at MM level (details in Suppl. Note. 4). All geometry optimizations were done while allowing the backbone and the hydrogen-bonded solvent molecules to relax to the QM solute (Medium level), while the rest of the MM water molecules are frozen (Low Layer). The ground state geometry minimum was obtained at the Møller−Plesset second order perturbation theory (MP2) level. All computations utilized the Pople 6-31 G* basis set employing the Cholesky decomposition method to speed up the computation of atomic integrals. The vertical energies and gradients for the LVC model were obtained at XMS-CASPT2 level utilizing an active space of 4 electrons in 4 orbitals, comprising the frontier HOMO and LUMO of the two bases. The validity of the minimal active space was established by benchmarking against larger active space computations discussed in Suppl. Note. 5. Structure analysis (NTOs, attachment and detachment densities, electron-hole distance) was performed with the WFA module of Molcas[77,78]. All CASPT2 computations were done with zero IPEA shift, and an imaginary shift of 0.2.

Transformation of the adiabatic XMS-CASPT2 states to locally-excited and pure-CT states was done by the Fragment Excitation Difference procedure outlined by Hsu et al.[79]. In this method an excitation matrix $\Delta x_{mn}$ is built which characterizes the excitation difference between the donor and acceptor. In our case, the diagonalization of this matrix separates out the locally excited states with eigenvalues +/−1 and the CT-state with an eigenvalue of 0. The transformation matrix thus obtained can be used to transform the Hamiltonian to the diabatic basis, to compute the static electronic coupling and the vibronically induced coupling between the diabatic states. Further details outlining the procedure are reported in Suppl. Note. 7.

ML-MCTDH dynamics were performed using a variant of the multilayer algorithm of Heidelberg MCTDH package[80] as implemented in Quantics[81] program. The system was parametrized with a Linear Vibronic Coupling model using energies, gradients and couplings computed at XMS-CASPT2 level. The wavepacket propagations were done with ML-MCTDH adopting a variable mean field scheme with a Runge-Kutta integrator up to a final time of 200 fs. The dynamics on different cluster representatives were performed with 62 photoactive vibrational modes selected from the total 84 normal modes of the

bichromophoric system included in the QM part of QM/MM setup. To select the photoactive modes, the maximum from the absolute value of either the gradients of the electronic states or interstate couplings of every pair of electronic states at Franck-Condon for each mode was chosen. All the 84 modes were sorted in decreasing order based on this value and the first 62 modes were selected. In this way, the discarded modes are the least active upon excitation as they have smallest gradients/interstate couplings at Franck-Condon (less than ~ 0.02 eV) and don't promote population transfer. The selected 62 modes were partitioned into system (16) and bath modes (46). For the primitive basis-set we adopted Hermite DVR functions. The 16 coordinates containing the largest coupling/gradients were grouped together by pairs of coordinates with similar frequency in the ML tree and described by a larger number of primitive (30) and single-particle functions, while the remainder 46 (the "bath") were grouped in pairs and described with less primitives (15) and single-particle functions. The 100 dynamics in different solvent ensembles were performed with 62 normal modes partitioned into a main system (8 modes) and bath (54 modes). The system modes were described 30 primitive bases and bath modes 15 primitive basis functions.

## Data availability

The QM/MM optimized structures for the clusters representatives employed in this work, the solvent configurations for the dynamics displayed in Fig. 2 and the parameters used for the Hamiltonian employed in the MLMCTDH wavepacket dynamics have been deposited in the Zenodo database under the accession code 10987710. Source data for Figs. 1–4 are provided in the Source Data file[82] (https://doi.org/10.6084/m9.figshare.25640532). Source data are provided with this paper.

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

## Acknowledgements

D.A. acknowledges Fundación Ramón Areces and Generalitat Valenciana/European Social Fund (APOSTD/2021/025) for fundings and ICCOM-CNR (Pisa)/ICMol-MolMatTC (Valencia) for hospitality. Support from the U.S. Department of Energy, Office of Science, Office of Basic Energy Sciences, Chemical Sciences, Geosciences and Biosciences Division under award no. DE-SC0022225 (A.N., F.M., S.M., V.K.J., and M.G.) is gratefully acknowledged. AN acknowledge financial support from PNRR MUR project ECS_00000033_ECOSISTER and from the Si-Fi-MYSTERY project, PRIN: PROGETTI DI RICERCA DI RILEVANTE INTERESSE NAZIONALE-Bando 2022, Prot. P2022WSJKS. G.C., F.S. and M.G. acknowledge financial support from the CRESCENDO project, PRIN: PROGETTI DI RICERCA DI RILEVANTE INTERESSE NAZIONALE-Bando 2022, Prot. 2022HL9PRP. G.C. and M.G. acknowledge financial support from the European Union – Next Generation EU – PNRR – M4C2, investimento 1.1 – "Fondo PRIN 2022" – "Understanding the pHotochemistry of sulfur substituted dnA bases by advanced ultrafast spectroscopy for phototherapeutic applications (HAPPY) ID P20224AWLB – CUP D53D23016720001".

## Author contributions

P.K. build the experimental setup, V.P., P.K., L.U. performed the measurements, V.P., P.K. analyzed and interpreted the experimental data. V.K.J, D.A., F.M., and S.U. performed the computational simulations. V.K.J., D.A., F.S., M.G., S.M., and A.N. analyzed and discussed the computational simulations. D.A., S.M., A.N., M.G., and G.C. procured funding for the work. F.S., A.N., M.G., and G.C. supervised the project. V.K.J. and A.N. wrote the manuscript with inputs from all authors.

## Competing interests

The authors declare no competing interests.
