## [Peer Review File · Nature Communications]

Sub-100-fs energy transfer in coenzyme NADH is a coherent process assisted by a charge-transfer stateReviewer #1 (Remarks to the Author):

In this study, the authors employ transient absorption spectroscopy with sub-30-fs temporal resolution to demonstrate that ultrafast energy transfer (EET) from adenine to nicotinamide in water-solvated NADH transpires within a 50-fs timeframe. The reported time constant, $\tau_{\text{EET}} = 54 \pm 11$ fs, seemingly implying a coherent process, is actually an amalgamation of multiple inherently coherent processes. These processes arise from the interplay between the dimer's conformational dynamics and the solvent's thermal fluctuations.

The authors assert that the EET process may be accelerated by involving a low-lying charge transfer (CT) state that interacts with the locally excited states of the two monomers. However, the claim about the involvement of the CT state lacks robust support from spectroscopic evidence, including vibrational or electronic signatures. Furthermore, the impact of solvent fluctuations should be more comprehensively addressed. Simply put, the proximity of the EET process's time constant to the solvent relaxation time of 100 fs does not convincingly elucidate the role of solvent fluctuations in EET.

The mechanisms governing energy transfer and charge separation processes are discussed in the context of relaxation times, which involve the interplay between electronic and nuclear degrees of freedom, conformational diversity, and solvent fluctuations. However, these arguments seem somewhat nebulous when applied to this specific system, making it challenging to draw definitive conclusions. In light of this, the manuscript falls short in proposing concrete strategies for optimizing EET efficiency, given the complex interplay of these factors.

In summary, this manuscript requires enhanced clarity and readability, achieved by simplifying the language and restructuring intricate sentences. Additionally, it should underscore the practical implications of this research for a broader audience.

Reviewer #2 (Remarks to the Author):

This paper possibly deserves a publication in Nature Communications.

The strength of this work is that it adopts an ab initio quantum approach to describe the dynamics of the system for a very large system in solvent.

This is not the first work of this kind, but, first, these state-of-the-art simulations are still very rare and here they are combined with an experiment.

(i) First, the context of the theoretical part must be improved. The authors gives several references for ML-MCTDH [43-49], but nothing about the historical works on MCTDH. ML-MCTDH is "just" an extension of MCTDH and the older references [first papers, review, book] on MCTDH should be given. If not, it gives the wrong feeling that everything started with Ref. [43] and they ignore that everything is based on the work of the group of Heidelberg.

For the vibronic coupling model, Refs. [47,48] must be corrected.

It is Köppel not Köuppel and I do not know why Ref.[48] is in capital letters.

They used the QUANTICS Package Ref.[63], but what they used is a variant (with secondary changes) of the Heidelberg package. This is not like for the DD-vMCG approach in QUANTICS. In particular, the ML-MCTDH algorithm they are using was developed in Heidelberg (Ref. [46]). They should say that they are using a variant of the Heidelberg MCTDH package (with the citation) as implemented in QUANTICS (with the reference).

They can simply look at the webpage: QUANTICS: Acknowledgments and citations.

Most importantly, they forget to cite some former works

that already developed a global strategy as the one in their paper: the most important reference being

"Multidimensional Quantum Mechanical Modeling of Electron Transfer and Electronic Coherence in Plant Cryptochromes : The Role of Initial Bath Conditions", The Journal of Physical Chemistry B 122 (2018) 126.

(ii) They claim that they performed a "full" nonadiabatic quantum dynamical simulation.

This is not true. They used 62 vibrational modes only.

First, they should justify very carefully the choice of these coordinates. I do not see such a justification even in the supplementary material.

To make a "full" quantum simulation, they should either include all the degrees of freedom or start with the spectrum they can obtain with the QM/MM simulations and then discretize the spectrum and increase the number of nuclear of freedom until it does not change the time constants. This can be called a "full" quantum simulation for the nuclei.

(iii) The most important and interesting point is the role of coherence.

There is a huge amount of publications and discussions about the role of quantum coherence in processes such as a charge-transfer in an environment.

However, this is not clear at all, if experimentalists observe quantum or classical coherence and if we have to deal with vibrational, electronic or vibronic quantum coherence.

The ab initio strategy adopted here is the correct way to give an answer to these questions according to me.

However, they did not calculate and discuss any quantum coherence !

This is not because they propagate wavepackets that the quantum interferences play a very important role in the process. They have to prove it.

I strongly suspect that they refer to quantum electronic coherence in this work.

They should calculate the quantum coherence between the different electronic states.

This is very easy and they do not need to make any new propagation.

If they do observe a quantum coherence that is not very small and that lasts

during a time that is not extremely small, this will be a very exciting result

and the authors will bring an important answer to the main question raised by the paper.

Reviewer #3 (Remarks to the Author):

The authors have studied the photoinduced electronic energy transfer from adenine to nicotinamide present in the reduced form of the NADH coenzyme. From the sub-30 fs time resolution in their experiments they conclude that the energy transfer step takes place with a time constant of 54 fs, which sets a much more precise value in comparison with previous studies which had indicated a sub-100 fs time scale. Overall, the study is robust in the experimental and computational sections. I will focus on the experimental results as the computational contribution should be reviewed by an expert of this area.

The main experimental result corresponds to the estimation of a 1/40 fs rate constant for the overall decay of the excited adenine chromophore for folded NADH molecules. The study obtains this rate constant considering a 70/30 unfolded/folded conformations (where approximately, only the folded isomers undergo energy transfer). The authors determine these rate constant for the folded isomers from the 122 fs components in the transient absorption experiments but taking into account the relative population where the folded conformation brings the two chromophores into the same solvation shell. From the overall rate constant and considering the unfolded case (in a solvent that does not favor the folded conformation), the time constant for energy transfer was determined to be 1/54 fs (which is actually within the experimental error of previous studies). This rapid energy transfer process was further studied by quantum dynamics simulations where a crucial role of the vibronic couplings was revealed (while in accordance with the experimental time - scale).

Overall, the study is robust in the experimental section which correspond to standard excited state

absorption experiments with sub 30 fs time resolutions. Certainly, the results are of significant value for a better understanding of the photophysics of cellular chromophores, and for a more precise understanding (possibly a bench-mark case) of coherent energy transfer processes.

From this, I recommend publication in Nature Communications after the following issues are considered by the authors.

1) Some recent research on the intrinsic photophysics of NADH should definitely be included:

J. Photochemistry and Photobiology A: Chemistry 2023 436, 114384

J. Phys. Chem. B 2020, 124, 5, 771–776

J. Phys. Chem. B 2023, 127, 39, 8432–8445

These recent studies have focused on early relaxation events on this kind of chromophores which need to be included. As the authors mention in page 5 at the end of the second paragraph, the rapid solvent response mentioned in these contributions could be relevant, the authors should make the respective connection (if appropriate) with the nicotinamide solvation results in water from the previous references.

2) Since the main result depends on the sorting out of the contribution of the folded isomers to the overall transient absorption 122 fs component, this treatment should be brought into the main manuscript from the Supporting Information (this is indeed the main experimental result and at least parts of sections 1.2 and 1.3 of the Supporting Infor should be discussed in the main text).

3) In page 5 in the second paragraph, the authors mention that EET is faster than IC since EET requires less pronounced structural deformations. The EET process is augmented significantly from the vibronic couplings but would not be absent otherwise, therefore, this sentence might cause some confusion for the reader (as it is being compared directly with the IC rate). The authors might consider here a writing where it is mentioned that the structural deformations related to EET “contribute” to the energy transfer rate being faster than the IC rate.

4) The absorbance of NADH at the excitation wavelength of these experiments is dominated by the absorption by Adenine, however, the direct absorption by the nicotinamide chromophore at this wavelength may not be insignificant. The authors should include more precise values for the relative absorbances by each chromophore at the excitation wavelength.

Reviewer #1

Reviewer's comment: In this study, the authors employ transient absorption spectroscopy with sub-30-fs temporal resolution to demonstrate that ultrafast energy transfer (EET) from adenine to nicotinamide in water-solvated NADH transpires within a 50-fs timeframe. The reported time constant, $\tau_{\text{EET}} = 54 \pm 11$ fs, seemingly implying a coherent process, is actually an amalgamation of multiple inherently coherent processes. These processes arise from the interplay between the dimer's conformational dynamics and the solvent's thermal fluctuations. The authors assert that the EET process may be accelerated by involving a low-lying charge transfer (CT) state that interacts with the locally excited states of the two monomers. However, the claim about the involvement of the CT state lacks robust support from spectroscopic evidence, including vibrational or electronic signatures.

Authors' reply: We agree with the Reviewer. Unfortunately, while ultrafast transient absorption (TA) spectroscopy probing in the UV/visible precisely quantifies the time constant for the EET process, it does not resolve the involvement of an intermediate CT state. We have simulated the TA spectrum and show that the CT state simply does not have strong enough PA features to allow its recognition. This makes its theoretical description much more relevant. We hope that our paper will stimulate further experimental work using different spectroscopic techniques and probe spectral windows that will allow us to identify the CT state and pinpoint its role in the EET process. In the revised manuscript we write:

“In particular, we propose photoelectron spectroscopy as a more sensitive way to single out CT states, as ionization from the intermediately created negatively charged Ade would give rise to characteristic signatures at lower ionization energies compared to the neutral form.”

Furthermore, the impact of solvent fluctuations should be more comprehensively addressed. Simply put, the proximity of the EET process's time constant to the solvent relaxation time of 100 fs does not convincingly elucidate the role of solvent fluctuations in EET.

Authors' reply: We respect the concerns of the Referee regarding the lack of modelling the dynamical solvent response in our wavepacket dynamics. However, we would like to highlight the major insights we have made in the work regarding the effect of solvent fluctuations on the EET process. We have made a very detailed analysis of the solvent and conformational effect on the energetics of the CT state at the time of excitation. We have shown how the different arrangements of solvent can lead to drastic changes in the relative energies of the CT state with respect to the LE states. We have also discussed how this dictates the participation of the CT state and influences the timescale of EET dynamics. Figures 3e and 3f in the main manuscript and the associated text detail this influence in two cases exemplifying the two different regimes of the solvent effect.

Reviewer's comment: The mechanisms governing energy transfer and charge separation processes are discussed in the context of relaxation times, which involve the interplay between electronic and nuclear degrees of freedom, conformational diversity, and solvent fluctuations. However, these arguments seem somewhat nebulous when applied to this specific system, making it challenging to draw definitive conclusions. In light of this, the manuscript falls short in proposing concrete strategies for optimizing EET efficiency, given the complex interplay of these factors.

Authors' reply: We understand the Referee's concerns that our work doesn't propose concrete strategies for optimizing EET in this system. However, we would like to emphasize that the

goal of this paper is not to optimize the EET efficiency in NADH, but rather to provide a comprehensive analysis of the reasons behind its sub-100-fs regime. We disentangled the individual effects of each factor (spatial proximity, stacking and solvent polarity), thus reflecting the complexity of the EET process even in a simple bi-chromophoric system and helping in a comprehensive understanding of the factors responsible for driving this process in ultrafast timescales. To better stress this point, we added the following text to the manuscript: “Our findings are of general validity for describing the dependence of energy transfer and charge separation processes on the coupling between electronic and nuclear degrees of freedom, conformational heterogeneity and solvent fluctuations in closely packed hetero-aggregates. The EET depends on the interplay of many factors: strength and nature of the tuning and coupling modes; conformational freedom and spatial distance between chromophores in the aggregate; solvent polarity and thermalization timescale. We identify the parameter window which facilitates ultrafast EET. At an inter-chromophore distance of $< 5\text{\AA}$: a) electronic and vibronic couplings are equally relevant and cooperate in promoting efficient EET; b) even localized vibrational modes are capable of generating sufficiently strong orbital mixing and, thus, strong vibronic couplings; c) the short distance between the centers of negative and positive charge of the two chromophores stabilizes the CT state opening an additional EET pathway, favored in polar solvents. Most importantly, despite the conformational heterogeneity of such macro-structures, the EET is an intrinsically coherent process, i.e. governed by coherently oscillating wave packets on the potential energy surfaces of the involved electronic states which tune the energy gaps and drive the EET unidirectionally from the donor to the acceptor. Such coherent EET cannot be described by Förster theory which is suited to describe incoherent EET mechanisms.”

Reviewer’s comment: In summary, this manuscript requires enhanced clarity and readability, achieved by simplifying the language and restructuring intricate sentences.

Authors’ reply: We appreciate the constructive criticism provided by the Referee and have taken to heart their advice. We have rewritten major portions of the manuscript to enhance clarity and readability. In Introduction, we have clarified the concepts of coherent and incoherent EET process to set the stage for our work which discusses ultrafast EET in a system with seemingly weakly coupled electronic states. We discuss how the relative energies and inter-chromophoric distances dictate the coupling between the electronic states to influence timescales of the EET process. In the main text we have made an effort to simplify our discussion while retaining the insights we have noted regarding the various factors which dictate the EET process in NADH. In the Results section, we detail the importance of both the static electronic and vibronic coupling responsible for the EET process. We highlight the factors responsible for controlling these mechanisms namely interbase distance, tuning and coupling modes and solvent configurations. We provide a version with Track Changes which highlights the modifications made.

Reviewer’s comment: Additionally, it should underscore the practical implications of this research for a broader audience.

Authors’ reply: We believe that our findings are general and hold for any aggregate, not just for NADH. We demonstrate that the EET process:

- a) Can be as fast ca. 50 fs and cannot be described by Förster theory, but is intrinsically coherent (i.e. governed by classical coherences which lead to recrossing between PES and efficient transfer);

- b) Is a complex process depending on the interplay of many factors: intramolecular distance (stacking); separation between centers of charge; geometrical deformations coupling donor and acceptor (the process does not have to involve large deformations; in fact, we identify the window within these parameters that promotes ultrafast EET);
- c) Can be accelerated by CT-mediated pathways under certain conditions which we clearly elucidate and relate to structural and environmental parameters.

We now extensively discuss these aspects in the revised version of the manuscript.

Reviewer #2

Reviewer's comment: This paper possibly deserves a publication in Nature Communications. The strength of this work is that it adopts an ab initio quantum approach to describe the dynamics of the system for a very large system in solvent. This is not the first work of this kind, but, first, these state-of-the-art simulations are still very rare and here they are combined with an experiment.

Authors' reply: We thank the Reviewer for the positive evaluation of our manuscript and for the constructive suggestions. We did our best to incorporate them in the revised manuscript.

Reviewer's comment: The authors gives several references for ML-MCTDH [43-49], but nothing about the historical works on MCTDH. ML-MCTDH is "just" an extension of MCTDH and the older references [first papers, review, book] on MCTDH should be given. If not, it gives the wrong feeling that everything started with Ref. [43] and they ignore that everything is based on the work of the group of Heidelberg.

Authors' reply: The appropriate references of the MCTDH have been added. An additional sentence has been added to the text.

Original version: "To gain insight into the ~50-fs intramolecular EET mechanism in the folded NADH forms, we have modeled the photoinduced time-evolution of the system by nonadiabatic quantum dynamics of multidimensional wavepackets using the ML-MCTDH method."

New version: "To gain insight into the ~50-fs Ade → Nic EET mechanism in the folded NADH forms, we have modeled the photoinduced time-evolution of the system by nonadiabatic quantum dynamics of multidimensional wavepackets using the MCTDH method.⁴⁸⁻⁵² Due to the large number of normal modes in the bichromophoric molecule, the multilayer formulation of the MCTDH method (ML-MCTDH) was employed.⁵³⁻⁵⁶"

Reviewer's comment: For the vibronic coupling model, Refs. [47,48] must be corrected. It is Köppel not Köuppel and I do not know why Ref.[48] is in capital letters.

Authors' reply: The spelling error is present in the online webpage and citation file of the manuscript from the journal website. However, indeed the correct spelling "Köppel" is used in the original manuscript. We thank the Reviewer for pointing this out and have corrected this error in our manuscript (reference [47] in revised manuscript). The case for reference [48] (new reference [58]) is fully capital in the official citation file and also the original published manuscript. So we have kept the text as is.

Reviewer's comment: They used the QUANTICS Package Ref.[63], but what they used is a

variant (with secondary changes) of the Heidelberg package. This is not like for the DD-vMCG approach in QUANTICS. In particular, the ML-MCTDH algorithm they are using was developed in Heidelberg (Ref. [46]). They should say that they are using a variant of the Heidelberg MCTDH package (with the citation) as implemented in QUANTICS (with the reference). They can simply look at the webpage: QUANTICS: Acknowledgments and citations.

Authors' reply: We have modified the text in the METHODS section and added the relevant citations.

Original version: "ML-MCTDH dynamics were performed by Quantics⁶⁹ program using Linear Vibronic Coupling model using energies, gradients and couplings computed at XMS-CASPT2 level."

New version: "ML-MCTDH dynamics were performed using a variant of the multilayer algorithm of Heidelberg MCTDH package^{56,77} as implemented in Quantics⁷⁸ program. The system was parametrized with a Linear Vibronic Coupling model using energies, gradients and couplings computed at XMS-CASPT2 level."

Reviewer's comment: Most importantly, they forget to cite some former works that already developed a global strategy as the one in their paper: the most important reference being "Multidimensional Quantum Mechanical Modeling of Electron Transfer and Electronic Coherence in Plant Cryptochromes: The Role of Initial Bath Conditions", The Journal of Physical Chemistry B 122 (2018) 126.

Authors' reply: We thank the Reviewer for pointing us out the relevant article. We have added the citation to the manuscript. We also added a few other studies that we found relevant to our work.

Ashkenazi, G.; Kosloff, R.; Ratner, M. A. Photoexcited Electron Transfer: Short-Time Dynamics and Turnover Control by Dephasing, Relaxation, and Mixing. *J. Am. Chem. Soc.* 1999, 121, 3386–3395.

Ando, K.; Sumi, H. Nonequilibrium Oscillatory Electron Transfer in Bacterial Photosynthesis. *J. Phys. Chem. B* 1998, 102, 10991–11000.

Xu, D.; Schulten, K. Coupling of protein motion to electron transfer in a photosynthetic reaction center: investigating the low temperature behavior in the framework of the spin–boson model. *Cem. Phys.* 1994, 182, 91–117.

Reviewer's comment: They claim that they performed a "full" nonadiabatic quantum dynamical simulation. This is not true. They used 62 vibrational modes only. First, they should justify very carefully the choice of these coordinates. I do not see such a justification even in the supplementary material. To make a "full" quantum simulation, they should either include all the degrees of freedom or start with the spectrum they can obtain with the QM/MM simulations and then discretize the spectrum and increase the number of nuclear of freedom until it does not change the time constants. This can be called a "full" quantum simulation for the nuclei.

Authors' reply: We acknowledge that this information was missing in the manuscript. We have added text in the METHODS section a description of the methodology used to select the most active photoactive modes. We analyzed the values of interstate couplings and gradients along

the 84 modes for various solvated conformers. These discarded modes were the ones having absolute value of gradient/interstate coupling below a threshold of 0.02 eV
We have also removed the “Full” word which was used once in the manuscript.

New version: “The dynamics on different cluster representatives were performed with 62 photoactive vibrational modes selected from the total 84 normal modes of the bichromophoric system included in the QM part of QM/MM setup. To select the photoactive modes, the maximum from the absolute value of either the gradients of the electronic states or interstate couplings of every pair of electronic states at Franck-Condon for each mode was chosen. All the 84 modes were sorted in decreasing order based on this value and the first 62 modes were selected. In this way, the discarded modes are the least active upon excitation as they have smallest gradients/interstate couplings at Franck-Condon (less than ~ 0.02 eV) and don't promote population transfer.”

Reviewer's comment: The most important and interesting point is the role of coherence. There is a huge amount of publications and discussions about the role of quantum coherence in processes such as a charge-transfer in an environment. However, this is not clear at all, if experimentalists observe quantum or classical coherence and if we have to deal with vibrational, electronic or vibronic quantum coherence. The ab initio strategy adopted here is the correct way to give an answer to these questions according to me. However, they did not calculate and discuss any quantum coherence! This is not because they propagate wavepackets that the quantum interferences play a very important role in the process. They have to prove it. I strongly suspect that they refer to quantum electronic coherence in this work. They should calculate the quantum coherence between the different electronic states. This is very easy and they do not need to make any new propagation. If they do observe a quantum coherence that is not very small and that lasts during a time that is not extremely small, this will be a very exciting result and the authors will bring an important answer to the main question raised by the paper.

Authors' reply: We thank the Reviewer for this stimulating comment. Following their suggestion we explored the role of the coherence. The analysis of the coherences made us realize that discussing several representative cases would allow also the general non-expert reader a better understanding of the non-adiabatic process.

Before we outline the analysis performed, we'd like to make clear that, unfortunately, no coherent features were found in the experimental spectrum. We believe that following reasons prohibit the experimental observation of coherences at this point:

- a) only 1/3 of the molecules are stacked and undergo EET, so that the signal at early times is dominated by signals from unstacked species;
- b) the sub-30-fs temporal resolution of the experiment does not allow to register modes with frequencies > 1000 cm⁻¹ thus the most important tuning and coupling mode (frequency 1600 cm⁻¹) in the direct EET mechanism, as well as in the Ade IC, cannot be resolved with the current experimental setup;
- c) the heterogeneity of the system is expected to accelerate decoherence in the ensemble of solvated molecules and to make it even more difficult to capture coherent features experimentally.

That being said, on the theoretical side we have access to the single snapshots which behave intrinsically coherently and our simulations do not suffer from the experimental limitations. As suggested by the Reviewer, we computed the quantum coherences between the three states La, Nic* and CT, involved in the EET for two representative cases discussed also in the main paper:

- a) the direct La->Nic* EET (Fig. 2f)
- b) the CT-mediated EET (Fig. 2d).

Below, we summarize the most relative findings.

We first discuss the direct EET (Figure R1 below, reported in the paper as Figure S18 in the SI). As expected, the effect of the CT state is vanishingly small, both on the populations and on the coherences. We observe a Nic*/La coherence (2nd row) lasting for nearly 70 fs, whose magnitude exhibits recurrences with times compatible to the periods of the most displaced normal modes with frequencies of 1400 cm⁻¹ and 1600 cm⁻¹ (about 20 fs period). As we discuss in the main text and with the help of figures Figure S12 and Figure S16 in SI, the coherence formation (and recurrence) is related to the passage through the La/Nic* CI region governed by the high-frequency tuning modes. In fact, degeneracy is reached every 10 fs which leads to population transfer and, simultaneously, coherence formation, as seen by the imaginary part of the Nic*/La coherence. The La→Nic* non-adiabatic transfer is facilitated both by electronic (labelled E_{ij}^0 in Figure 1) and vibronic (λ_{ij}) couplings which can be verified switching off either of the two contributions selectively (2nd and 3rd columns). The population transfer becomes slower in both cases (still leading to population inversion within 100 fs) and the coherence lasts for a longer time. While we already supplied Supplementary Figures (Figures S13-17) showing the effect of selectively switching off these terms on the overall population dynamics, we agree and thank the Reviewer that the analysis of coherences provides additional insights into the underlying mechanism.

Figure R1 Evolution of coherences and populations in the case of direct EET.

Next, we look at the CT-mediated EET (Figure R2). The La-CT coherence is built up almost instantaneously, despite La and CT having a gap of 0.5 eV in the FC point, as the gap is circumvented in few fs along the high-frequency tuning modes. Both electronic and vibronic couplings – exhibiting larger magnitudes compared to their counterparts in the direct EET due to the one-electron nature of the process – facilitate the formation of the coherence which can

be seen when switching them off selectively (2nd and 3rd columns). Also in this example, the coherence shows recurrences with times compatible with the periods of the most displaced normal modes.

The CT/La coherence decays in about 70 fs (3rd row), at the same time that a coherence builds up between CT and Nic* and lasts till about 175 fs (1st row). The temporal profile of the Nic*/CT coherence is less structured.

If we renormalize coherences between states by $1/\sqrt{p_i p_j}$ (where p_i and p_j are the populations in the i -th and j -th states) we still see a very fast decay of the CT/La coherence, which indicates that the decoherence is due to the decrease of the overlap of the nuclear wave packets evolving on the two surfaces (magenta plots in both Figures). The decoherence is accentuated by the vibronic couplings, when these are removed, we observe a higher degree of nuclear coherence (Figure 2 below, 3rd column, magenta plot of CT/La coherence).

Figure R2 Evolution of coherences and populations in the case of CT-mediated EET in which the La and CT state have a 0.5 eV gap in the FC point.

As it can be seen from the analysis of the two snapshots, the role of the tuning modes is essential for facilitating EET. If we switch them off (labelled λ_{ii} , 5th column in Figures R1 and R2), we observe high frequency oscillations in the coherences proportional to the electronic energy gaps due to the purely electronic coupling E_{ij}^0 . Despite the coherences having sizeable values, no effective population transfer takes place. Hence, despite the conformational heterogeneity of NADH, the EET is an intrinsically coherent process governed by coherently oscillating wave packets on the potential energy surfaces of the involved electronic states which tune the energy gaps and drive the EET unidirectionally from the donor to the acceptor. Such coherent EET cannot be described by incoherent mechanisms such a Förster theory.

Experimentally, one could expect to see quantum beating in the transient absorption spectra, manifestations of the classical coherences due to the coherent vibrational motion of the wave packet on the harmonic PES. Despite having computed the coherence only for a few snapshots we are certain that the same high frequency tuning modes will be activated in all snapshots. Moreover, the classical coherence will be observed also in unstacked samples in which Adenine undergoes IC. Unfortunately, at this point, the high frequencies of the dominant modes prevent them from being registered with the available experimental setup.

On a sidenote, transient absorption is not a suitable method for registering electronic coherences, as the contributions are covered by the more intense population contributions. More advanced spectroscopies such as TRUECARS (M Kowalewski, K. Bennett, K.E. Dorfman, and S. Mukamel, Phys. Rev. Lett. 115, 193003 (2015)) have been designed to resolve specifically electronic coherences.

The above discussion was incorporated in the revised draft. Figures R1-R2 presented here were added in the SI as Figures S18-S19.

Reviewer #3

Reviewer's comment: The authors have studied the photoinduced electronic energy transfer from adenine to nicotinamide present in the reduced form of the NADH coenzyme. From the sub-30 fs time resolution in their experiments they conclude that the energy transfer step takes place with a time constant of 54 fs, which sets a much more precise value in comparison with previous studies which had indicated a sub-100 fs time scale. Overall, the study is robust in the experimental and computational sections. I will focus on the experimental results as the computational contribution should be reviewed by an expert of this area.

The main experimental result corresponds to the estimation of a 1/40 fs rate constant for the

Figure R3 Evolution of coherences and populations in the case of CT-mediated EET in which the La and CT state are degenerate in the FC point

overall decay of the excited adenine chromophore for folded NADH molecules. The study obtains this rate constant considering a 70/30 unfolded/folded conformations (where approximately, only the folded isomers undergo energy transfer). The authors determine these rate constant for the folded isomers from the 122 fs components in the transient absorption experiments but taking into account the relative population where the folded conformation brings the two chromophores into the same solvation shell. From the overall rate constant and considering the unfolded case (in a solvent that does not favor the folded conformation), the time constant for energy transfer was determined to be 1/54 fs (which is actually within the experimental error of previous studies). This rapid energy transfer process was further studied by quantum dynamics simulations where a crucial role of the vibronic couplings was revealed (while in accordance with the experimental time – scale).

Overall, the study is robust in the experimental section which correspond to standard excited state absorption experiments with sub 30 fs time resolutions. Certainly, the results are of significant value for a better understanding of the photophysics of cellular chromophores, and for a more precise understanding (possibly a bench-mark case) of coherent energy transfer processes.

From this, I recommend publication in Nature Communications after the following issues are considered by the authors.

Authors' reply: We thank the Reviewer for the positive evaluation of our work and for considering our results “robust” and potentially setting a “bench-mark case” for the understanding of coherent energy transfer processes. In the following, we fully address their comments.

Reviewer's comment: Some recent research on the intrinsic photophysics of NADH should definitely be included:

J. Photochemistry and Photobiology A: Chemistry 2023 436, 114384

J. Phys. Chem. B 2020, 124, 5, 771–776

J. Phys. Chem. B 2023, 127, 39, 8432–8445

These recent studies have focused on early relaxation events on this kind of chromophores which need to be included. As the authors mention in page 5 at the end of the second paragraph, the rapid solvent response mentioned in these contributions could be relevant, the authors should make the respective connection (if appropriate) with the nicotinamide solvation results in water from the previous references.

Authors' reply: We thank the Reviewer for pointing out these works to us. While the above works do discuss solvation response, they focus on fluorescence response pertinent to the solvent relaxation around the excited Nic* chromophore in NADH/isolated Nic monomer. In the context of our work, these are processes occurring after the EET process is completed, and relevant to relaxation of the excited Nic* chromophore. However, these works are indeed relevant in the broader context of NADH photophysics and therefore we have included them in the Introduction of the revised manuscript as Refs. 31-33.

Reviewer's comment: Since the main result depends on the sorting out of the contribution of the folded isomers to the overall transient absorption 122 fs component, this treatment should be brought into the main manuscript from the Supporting Information (this is indeed the main experimental result and at least parts of sections 1.2 and 1.3 of the Supporting Infor should be discussed in the main text).

Authors' reply: We agree with the Reviewer that this is an important discussion which should be brought into the main text. We have moved the Table S1 of SI containing the lifetimes obtained by Global Fitting of the TA spectra to the main text as Table 1. We have also included a concise version of the discussion about the computation of EET rates in the main text. The revised text reads:

“Global fitting of the TA data, as shown in Table 1, gives time constants of $\tau_{\text{Ade}} = 157 \pm 4$ fs and $\tau_{\text{NADH, water}} = 122 \pm 4$ fs for Ade and NADH in water, respectively. While in isolated Ade and NADH solvated in methanol, the corresponding time constants of 157 fs and 162 fs correspond to the IC back to the GS, in NADH solvated in water the EET mechanism opens up an additional channel of excited state deactivation, increasing its rate. Taking into account the 30/70 equilibrium between folded and unfolded forms, the faster lifetime $\tau_{\text{NADH, water}} = 122$ fs extracted by global analysis can be considered as a weighted average of the folded and unfolded ultrafast lifetime components:

$$\tau_{\text{NADH, water}} = 0.7 \tau_{\text{NADH, unfolded}} + 0.3 \tau_{\text{NADH, folded}}$$

Assuming similar excited-state deactivations in unfolded water solvated NADH and isolated Ade, one can take $\tau_{\text{NADH, unfolded}} = \tau_{\text{Ade, water}} = 157$ fs, which results in a time constant $\tau_{\text{NADH, folded}} = 40 \pm 6$ fs for the folded NADH conformers. This total decay rate obtained for the folded NADH components ($k_{\text{total}} = (40 \text{ fs})^{-1}$) reflects the sum of the rates for Ade IC ($k_{\text{IC}} = (157 \text{ fs})^{-1}$) and ultrafast EET from the Ade La-state to Nic* (k_{ET}), resulting in $k_{\text{total}} = k_{\text{IC}} + k_{\text{ET}}$, thus

giving a time constant of $\tau_{\text{ET, folded}} = 54 \pm 11$ fs for the EET process (see a detailed discussion in Sections 1.2 and 1.3 of the SI). “

Reviewer’s comment: In page 5 in the second paragraph, the authors mention that EET is faster than IC since EET requires less pronounced structural deformations. The EET process is augmented significantly from the vibronic couplings but would not be absent otherwise, therefore, this sentence might cause some confusion for the reader (as it is being compared directly with the IC rate). The authors might consider here a writing where it is mentioned that the structural deformations related to EET “contribute” to the energy transfer rate being faster than the IC rate.

Authors’ reply: We agree with the point raised by the Reviewer. We have modified the sentence which now reads:

“Through multidimensional nonadiabatic quantum dynamics, we demonstrate that this ultrafast EET in folded conformers is channeled through coherent vibrational motion along the same planar modes that facilitate intra-Ade IC. These coherent molecular vibrations contribute to the EET being faster than IC to the GS as for the latter process additional out-of-plane structural deformations are needed to bring the La and GS electronic potential energy surfaces to degeneracy.”

Reviewer’s comment: The absorbance of NADH at the excitation wavelength of these experiments is dominated by the absorption by Adenine, however, the direct absorption by the nicotinamide chromophore at this wavelength may not be insignificant. The authors should include more precise values for the relative absorbances by each chromophore at the excitation wavelength.

Authors’ reply:

We express our gratitude to the reviewer for providing an opportunity to clarify this aspect and improve the clarity of our experimental section.

Considering the reported analogues of Nicotinamide (Nic) found in the literature (Cadena-Caicedo, A., Gonzalez-Cano, B., López-Arteaga, R., Esturau-Escofet, N. & Peon, J. Ultrafast Fluorescence Signals from β -Dihydropyridinone Adenine Dinucleotide: Resonant Energy Transfer in the Folded and Unfolded Forms. *J. Phys. Chem. B* **124**, 519–530 (2020); Reza, M. M. et al. Primary Photophysics of Nicotinamide Chromophores in Their Oxidized and Reduced Forms. *J. Phys. Chem. B* **127**, 8432–8445 (2023)), our excitation pulse at 4.7 eV coincides with the region close to the trough of the absorption spectrum of Nic. To offer a quantitative assessment, the absorption of Nic at 4.7 eV is ≈ 12 times lower in amplitude than its absorption peak around 3.6 eV. Factoring in the relative extinction coefficient of Adenine (Ade) in water at 4.7 eV ($\approx 13000 \text{ cm}^{-1} \text{ M}^{-1}$; Fasman, G. D., Editor (1975) Handbook of Biochemistry and Molecular Biology, 3rd Edition, Nucleic Acids, Volume I, pp. 65-215, CRC Press, Cleveland, Ohio.) compared to that of NADH in water at 3.6 eV ($\approx 6300 \text{ cm}^{-1} \text{ M}^{-1}$; McComb RB, Bond LW, Burnett RW, Keech RC, Bowers GN Jr. Determination of the molar absorptivity of NADH. *Clin Chem.* 1976 Feb;22(2):141-50. PMID: 2388.; Osik, N.A., Zelentsova, E.A., Sharshov, K.A. et al. Nicotinamide adenine dinucleotide reduced (NADH) is a natural UV filter of certain bird lens. *Sci Rep* **12**, 16850 (2022)), which is approximately 2, the anticipated direct excitation of the Nic moiety is expected to be around 1/24 of the contribution of the Ade absorption at 4.7 eV. As a result, the 2 OD absorption at 4.7 eV in our experiment corresponds to 1.92 OD of Ade and 0.08 OD of Nic absorptions, respectively. Consequently, the direct absorption of Nic amounts to approximately 4% when resonantly exciting the Ade moiety.

Certainly, as suggested by the reviewer, despite the small percentage of direct Nic excitation, it could have a notable impact on the dynamics. However, it's crucial to emphasize that the excitation energy transfer (EET) mechanism occurs coherently, as extensively detailed in the main text, and at a rate way faster than the vibrational cooling to the minima of the potential energy surfaces of the monomeric moieties. Studies on NADH or Nic moieties indicate that vibrational relaxation to the Nic minimum occurs in mere picoseconds (Cao, S. et al. Femtosecond Fluorescence Spectra of NADH in Solution: Ultrafast Solvation Dynamics. *J. Phys. Chem. B* 124, 771–776 (2020).; Heiner, Z., Roland, T., Leonard, J., Haacke, S. & Groma, G. I. Kinetics of Light-Induced Intramolecular Energy Transfer in Different Conformational States of NADH. *J. Phys. Chem. B* 121, 8037–8045 (2017).), at least one order of magnitude slower than the EET itself. This process may only influence the longer time constants of our global analysis, namely, the τ_2 and τ_3 components. Thus, the τ_1 time constant of NADH in water exclusively illustrates the additional deactivation channel found in Ade due to the presence of the EET. Therefore, the master equation:

$$\tau_{\text{NADH, water}} = 0.7 \tau_{\text{NADH, unfolded}} + 0.3 \tau_{\text{NADH, folded}}$$

is exclusively describing the sub-170-fs processes embedded in the τ_1 time constant and it is not affected by the direct population of 4% of the Nic moiety.

Taking this opportunity, we have incorporated the above calculations and line of thought into the experimental section of the supporting information for clarification.

We add the following text in the section 1.3 of the SI:

“However, we should stress the 4.7 eV resonant excitation of the Ade moiety, might result in concurrent direct population of a small percentage of the Nic moiety. Taking into account that: (i) the absorption of Nic at 4.7 eV is ≈ 12 times lower in amplitude than its absorption peak around 3.6 eV^{3,4}(ii) the extinction coefficient of Ade in water at 4.7 eV ($\approx 13000 \text{ cm}^{-1} \text{ M}^{-1}$)⁵ (iii) the extinction coefficient of NADH in water at 3.6 eV ($\approx 6300 \text{ cm}^{-1} \text{ M}^{-1}$)^{6,7}; the anticipated direct excitation of the Nic moiety is expected to be around 1/24 of the contribution of the Ade absorption at 4.7 eV. As a result, the 2 OD absorption at 4.7 eV in our experiment corresponds to 1.92 OD of Ade and 0.08 OD of Nic absorptions, respectively.

Despite the 4% of direct Nic population upon 4.7 eV excitation, the vibrational relaxation to the Nic minimum occurs in mere picoseconds^{1,2}, at least one order of magnitude slower than the EET itself. This process may only influence the longer time constants of our global analysis, namely, the τ_2 and τ_3 components of Table S1. Thus, the τ_1 time constant of NADH in water exclusively illustrates the additional deactivation channel found in Ade due to the presence of the EET, being unaffected by the small percentage (4%) of Nic direct excitation and without influencing the result of our master equation Eq.1.”

Reviewer #1 (Remarks to the Author):

The authors elucidate the correlation between the relative energies and inter-chromophoric distances, emphasizing their role in determining the coupling strength between electronic states, consequently influencing the timescales of the Energy Transfer (EET) process. Throughout the main text, a deliberate effort has been undertaken to streamline the discourse while preserving key insights into the diverse factors governing the EET process in NADH. The Results section extensively delves into the significance of both static electronic and vibronic coupling, elucidating their respective contributions to the EET process. The authors underscore the pivotal factors regulating these mechanisms, including interbase distance, tuning and coupling modes, and solvent configurations. Notably, the authors have incorporated Track Changes to meticulously document the modifications made, ensuring transparency in the evolution of their scientific narrative.

Reviewer #2 (Remarks to the Author):

The role of quantum coherence in non-adiabatic processes in very large systems, including biological systems, has been the subject of many intense debates and is still an open question.

The authors have replied to all my comments and questions correctly. I appreciate the fact that they made the effort to calculate the quantum coherence. They do see some quantum coherence in the calculations, but not in the experiments. I understand their arguments explaining why the quantum coherence is not observed in the experiments. The paper deserves publication now.

I have just one last comment. They explain that due to the experimental conditions, the quantum coherence is dissipated very quickly, but that the quantum coherence is present as shown by the calculations. This is perhaps true, but this is not totally sure: only an agreement between theory and experiment on the impact of the quantum coherence could provide a decisive answer.

If they could improve the experimental conditions in the future, to my opinion, this could deserve publications in Nature Chemistry or even Nature since the methodology that they have adopted is probably the only one that can extract the information about this quantum coherence.

I just think that they are a little bit too peremptory in their claim that the process is coherent, in particular in the beginning of the paper. They should probably qualify their statements.

Reviewer #3 (Remarks to the Author):

In the new version of the manuscript (Feb. 2024), the authors have appropriately addressed the issues related to the direct experimental observations made by this reviewer for the previous version of the text. The experimental work is now clear and there are more points of reference to previous studies. With regards to the ab initio quantum approaches, referees 1 and 2 appear to have more experience than this reviewer so I consider that their comments and requirements of the adjustment of the manuscript should be considered by each of them.

In my opinion the manuscript can be published in its current version.

Reviewer #1

Reviewer's comment: The authors elucidate the correlation between the relative energies and inter-chromophoric distances, emphasizing their role in determining the coupling strength between electronic states, consequently influencing the timescales of the Energy Transfer (EET) process. Throughout the main text, a deliberate effort has been undertaken to streamline the discourse while preserving key insights into the diverse factors governing the EET process in NADH. The Results section extensively delves into the significance of both static electronic and vibronic coupling, elucidating their respective contributions to the EET process. The authors underscore the pivotal factors regulating these mechanisms, including interbase distance, tuning and coupling modes, and solvent configurations. Notably, the authors have incorporated Track Changes to meticulously document the modifications made, ensuring transparency in the evolution of their scientific narrative.

Authors' reply: We thank the referee for his/her positive comments. No further actions are required.

Reviewer #2

Reviewer's comment: The role of quantum coherence in non-adiabatic processes in very large systems, including biological systems, has been the subject of many intense debates and is still an open question.

The authors have replied to all my comments and questions correctly. I appreciate the fact that they made the effort to calculate the quantum coherence. They do see some quantum coherence in the calculations, but not in the experiments. I understand their arguments explaining why the quantum coherence is not observed in the experiments. The paper deserves publication now.

Authors' reply: We thank the referee for recommending publication.

I have just one last comment. They explain that due to the experimental conditions, the quantum coherence is dissipated very quickly, but that the quantum coherence is present as shown by the calculations. This is perhaps true, but this is not totally sure: only an agreement between theory and experiment on the impact of the quantum coherence could provide a decisive answer.

If they could improve the experimental conditions in the future, to my opinion, this could deserve publications in Nature Chemistry or even Nature since the methodology that they have adopted is probably the only one that can extract the information about this quantum coherence. I just think that they are a little bit too peremptory in their claim that the process is coherent, in particular in the beginning of the paper. They should probably qualify their statements.

Authors' reply: We thank the reviewer for the positive evaluation of our manuscript and its scientific values. We agree with his/her suggestions, and we did our best to incorporate them in the revised manuscript. In particular, we performed the listed modifications to smooth down the peremptory effect of our statements:

In Abstract:

Original sentence: Nonadiabatic quantum dynamical simulations computed through the time-evolution of multidimensional wavepackets *reveal* that the population transfer is mediated by photoexcited molecular vibrations due to strong coupling between the electronic states.

New Sentence: Nonadiabatic quantum dynamical simulations computed through the time-evolution of multidimensional wavepackets *suggest* that the population transfer is mediated by photoexcited molecular vibrations due to strong coupling between the electronic states.

In Introduction:

Original Sentence: Through multidimensional nonadiabatic quantum dynamics, we *demonstrate* that this ultrafast EET in folded conformers is channeled through coherent vibrational motion along the same planar modes that facilitate intra-Ade IC.

New Sentence: Through multidimensional nonadiabatic quantum dynamics, we *anticipate* that this ultrafast EET in folded conformers is channeled through coherent vibrational motion along the same planar modes that facilitate intra-Ade IC.

Reviewer #3

Reviewer's comment: In the new version of the manuscript (feb. 2024), the authors have appropriately addressed the issues related to the direct experimental observations made by this reviewer for the previous version of the text. The experimental work is now clear and there are more points of reference to previous studies. With regards to the ab initio quantum approaches, referees 1 and 2 appear to have more experience than this reviewer so I consider that their comments and requirements of the adjustment of the manuscript should be considered by each of them.

In my opinion the manuscript can be published in its current version.

Authors' reply: We thank the Reviewer for the positive evaluation of our work and for considering our manuscript ready for publication in its current form.